# Fast and Efficient Matching Algorithm with Deadline Instances

Zhao Song[1], Weixin Wang[2], Chenbo Yin[3], Junze Yin[4]

[1]Simons Institute for the Theory of Computing, UC Berkeley, [2]Johns Hopkins University, [3]University of Texas at Austin, [4]Rice University

`magic.linuxkde@gmail.com, wwang176@jh.edu, chenboyin1@gmail.com, jy158@rice.edu`

The online weighted matching problem is a fundamental problem in machine learning due to its numerous applications. Despite many efforts in this area, existing algorithms are either too slow or don't take deadline (the longest time a node can be matched) into account. In this paper, we introduce a market model with deadline first. Next, we present our two optimized algorithms (FastGreedy and FastPostponedGreedy) and offer theoretical proof of the time complexity and correctness of our algorithms. In FastGreedy algorithm, we have already known if a node is a buyer or a seller. But in FastPostponedGreedy algorithm, the status of each node is unknown at first. Then, we generalize a sketching matrix to run the original and our algorithms on both real data sets and synthetic data sets. Let $\epsilon \in (0, 0.1)$ denote the relative error of the real weight of each edge. The competitive ratio of original Greedy and PostponedGreedy is $\frac{1}{2}$ and $\frac{1}{4}$ respectively. Based on these two original algorithms, we proposed FastGreedy and FastPostponedGreedy algorithms and the competitive ratio of them is $\frac{1-\epsilon}{2}$ and $\frac{1-\epsilon}{4}$ respectively. At the same time, our algorithms run faster than the original two algorithms. Given $n$ nodes in $\mathbb{R}^d$, we decrease the time complexity from $O(nd)$ to $\widetilde{O}(\epsilon^{-2} \cdot (n + d))$, where for any function $f$, we use $\widetilde{O}(f)$ to denote $f \cdot \mathrm{poly}(\log f)$.

## 1. Introduction

The online weighted matching problem is a fundamental problem with numerous applications, e.g. matching jobs to new graduates [1], matching customers to commodities [2], matching users to ads [3]. Motivated by these applications, researchers have spent decades designing algorithms to solve this problem [4–13]. Let $n$ denote the number of items that can be very large in real-world settings. For example, TikTok pushes billions of advertisements every day. Thus, it is necessary to find a method to solve this matching problem quickly. Finding the maximum weight is one of the processes of this method. Our goal is to provide faster algorithms to solve the online weighted matching problem with deadline by optimizing the part of finding the maximum weight. In a real-world setting, the weight of the edge between each pair of nodes can show the relationship between two nodes (for example, the higher the weight is, the more likely the buyer wants to buy the product). Each node may have multiple entries to describe its attribute, such as the times of searching for a specific kind of merchandise. And there might exist a time limit called deadline to confine the longest time a node can be matched. For example, most meat and milk in Amazon need to be sold before it goes bad. This shows the significance of solving the online weighted matching problem with deadline. Let $d$ be the original node dimension and dl denote deadline. Sometimes the weight is already provided, but we offer a new method to calculate the weight of the edge. This helps accelerate the process of finding the maximum weight.

In this paper, we solve the online matching problem with deadline and inner product weight matching. We introduce a sketching matrix to calculate the weight of the edge. By multiplying a vector with this matrix, we can hugely decrease the number of entries. Thus, it takes much less time to calculate the norm of the transformed vector. For $n$ nodes with $d$ entries each, we decrease the time complexity from $O(nd)$ to $\widetilde{O}(\epsilon^{-2} \cdot (n + d))$. In our experiment part, we also prove that the total matching value of

our FASTGREEDY and FASTPOSTPONEDGREEDY is very close to GREEDY and POSTPONEDGREEDY in practice, which means our approximated weight is very similar to the real weight.

## 1.1. Related Work

**Online Weighted Bipartite Matching.** Online weighted bipartite matching is a fundamental problem that has been studied by many researchers. [14] provided basic and classical algorithms to solve this problem. Our experimental results demonstrate that our optimized version achieves 10-20x speedup while maintaining similar matching quality, making it valuable for time-constrained matching scenarios, like [6, 10, 13, 15–19].

**Fast Algorithm via Data Structure.** Over the last few years, solving many optimization problems has boiled down to designing efficient data structures. For example, linear programming [20–25], empirical risk minimization [26, 27], cutting plane method [28], computing John Ellipsoid [29, 30], exponential and softmax regression [31–33], integral minimization problem [34], matrix completion [35], discrepancy [36–38], training over-parameterized neural tangent kernel regression [38–42], matrix sensing [43, 44].

**Roadmap.** In Section 2, we introduce the model we use. We provide our two optimized algorithms and their proof of correctness in Section 3. We use experiments to justify the advantages and the correctness of our algorithms in Section 4. At last, we draw our conclusion in Section 5.

## 2. Preliminaries

**Notation.** We use $\| \cdot \|_2$ to denote $\ell_2$ norm. For any function $f$, we use $\widetilde{O}(f)$ to denote $f \cdot \mathrm{poly}(\log f)$, where $\mathrm{poly}(\log f)$ refers to the polynomial of $\log f$. For integer $n$, we use $[n]$ to denote $\{1, 2, \ldots, n\}$. For a set $S$, we use $|S|$ to denote its cardinality.

## 2.1. Model

Now, given a bipartite graph $G$, we start by defining matching.

**Definition 2.1.** *Let $G = (V_1, V_2, E)$ denote a bipartite graph with $|V_1| = |V_2|$. We say $S \subset E$ is a matching if $|S| = |V_1|$, for each vertex $v \in V_1$ there is exactly one edge $e$ in $S$ such that $v$ is one of the vertexes in $e$, and for each vertex $u \in V_2$ there is exactly one edge $e$ in $S$ such that $u$ is one of the vertexes in $e$. Let $w : E \to \mathbb{R}$ denote a weight function. Let $w_e$ denote the weight of edge $e \in E$. We say $w(S) = \sum_{e \in S} w_e$ is the weight of matching $S$. Our goal is to make $w(S)$ as large as possible.*

**Definition 2.2.** *Let $[n]$ denote $\{1, \ldots, n\}$. Let $\mathcal{S}$ denote the set of all the sellers. Let $\mathcal{B}$ denote the set of all the buyers. Each set contains $n$ vertices indexed by $i \in [n]$. Each node arrives sequentially time $t \in [n]$. And for a seller $s \in \mathcal{S}$, it can only be matched in $\mathrm{dl}$ time after it reaches the market.*

**Definition 2.3.** *We create an undirected bipartite graph $G(\mathcal{S}, \mathcal{B}, E)$. We let $v_{i,j} \geq 0$ denote the weight of edge $e \in E$ between node $i$ and node $j$.*

**Definition 2.4.** *Let $m : \mathcal{S} \to \mathcal{B}$ denote a matching function. For a seller $s \in \mathcal{S}$, there is a buyer $b \in \mathcal{B}$ matched with seller $s$ if $m(s) = b$.*

## 2.2. Useful Lemma

JL Lemma creates a sketching matrix to accelerate the process of calculating the distance between two points.

**Lemma 2.5** (JL Lemma, [45])**.** *For any $X \subset \mathbb{R}^d$ of size $n$, there exists an embedding $f : \mathbb{R}^d \to \mathbb{R}^s$ where $s = O(\epsilon^{-2} \log n)$ such that $(1 - \epsilon) \cdot \|x - y\|_2 \leq \|f(x) - f(y)\|_2 \leq (1 + \epsilon) \cdot \|x - y\|_2$, where $x, y \in X$.*

## 3. Algorithm

In this section, we present the important properties of the algorithms.

**Lemma 3.1** (Restatement of Lemma A.1). *STANDARDGREEDY in Algorithm 5 is a $1/2$-competitive algorithm.*

**Lemma 3.2** (Restatement of Lemma A.2). *Let $w_{i,j}$ denote the real weight of the edge between seller $i$ and buyer $j$, and $\widetilde{w}_{i,j}$ denote the approximated weight. Let $\epsilon \in (0, 0.1)$ denote the precision parameter. If for all $\epsilon$, there exists an $\alpha$-approximation algorithm for the online weighted matching problem and a $\delta > 0$, where $(1-\delta)w_{i,j} \leq \widetilde{w}_{i,j} \leq (1+\delta)w_{i,j}$ for any seller node $i$ and buyer node $j$, there exists a greedy algorithm with competitive ratio $\alpha(1-\epsilon)$.*

**Lemma 3.3** (Restatement of Lemma A.3). *If a node is determined to be a seller or a buyer with $1/2$ probability in STANDARDGREEDY in Algorithm 5, then this new STANDARDPOSTPONEDGREEDY algorithm is a $1/4$-competitive algorithm.*

**Theorem 3.4.** *For precision parameter $\epsilon \in (0, 0.1)$, FASTGREEDY is a $\frac{1-\epsilon}{2}$-competitive algorithm.*

*Proof.* According to Lemma 3.1, there exists a $\frac{1}{2}$-competitive algorithm. According to Lemma 2.5, we create a sketching matrix $M \in \mathbb{R}^{s \times d}$, let $f(x) = Mx$ and $w_i = \|y_j - x_i\|_2$ denote the real matching weight of the edge between seller node $x_i$ and buyer node $y_j$, and $\widetilde{w}_{i,j} = \|f(y_j) - f(x_i)\|_2$ denote the approximated weight of the edge between seller node $x_i$ and buyer node $y_j$, then there will be $(1-\epsilon)w_{i,j} \leq \widetilde{w}_{i,j} \leq (1+\epsilon)w_{i,j}$ for $\forall i, j \in [n]$. Then according to Lemma 3.2, we can conclude that the competitive ratio of FASTGREEDY is $\frac{1-\epsilon}{2}$. $\qquad\square$

**Theorem 3.5** (Restatement of Theorem A.4, Correctness of FASTPOSTPONEDGREEDY in Algorithm 3 and Algorithm 4). *FASTPOSTPONEDGREEDY is a $\frac{1-\epsilon}{4}$-competitive algorithm.*

**Theorem 3.6.** *Consider the online bipartite matching problem with the inner product weight, for any $\epsilon \in (0, 1)$, $\delta \in (0, 1)$, there exists a data structure FASTGREEDY that uses $O(nd + \epsilon^{-2}(n+d)\log(n/\delta))$ space. FASTGREEDY supports the following operations*

- *INIT($\{x_1, x_2, \ldots, x_n\} \subseteq \mathbb{R}^d, \epsilon \in (0, 1), \delta \in (0, 1)$). Given a series of offline points $x_1, x_2, \ldots, x_n$, a precision parameter $\epsilon$ and a failure tolerance $\delta$ as inputs, this data structure preprocesses in $O(\epsilon^{-2}nd\log(n/\delta))$*

- *UPDATE($y \in \mathbb{R}^d$). It takes an online buyer point $y$ as inputs and runs in $O(\epsilon^{-2}(n+d)\log(n/\delta))$ time. Let $t \in \mathcal{N}$ denote the arrival time of any online point.*

- *QUERY($y \in \mathbb{R}^d$). Given a query point $y \in \mathbb{R}^d$, the QUERY approximately estimates the Euclidean distances from $y$ to all the data points $x_1, x_2, \ldots, x_n \in R^d$ in time $O(\epsilon^{-2}(n+d)\log(n/\delta))$. For $\forall i \in [n]$, it provides estimates estimates $\{\widetilde{w}_i\}_{i=1}^n$ such that: $(1-\epsilon)\|y-x_i\|_2 \leq \widetilde{w}_i \leq (1+\epsilon)\|y-x_i\|_2$.*

- *TOTALWEIGHT. It outputs the matching value between offline points and $n$ known online points in $O(1)$ time. It has competitive ratio $\frac{1-\epsilon}{2}$ with probability at least $1 - \delta$.*

We prove the data structure (see Algorithm 1 and Algorithm 2) satisfies the requirements of Theorem 3.6 by proving the following lemmas.

**Lemma 3.7.** *The procedure INIT (Algorithm 1) in Theorem 3.6 runs in time $O(\epsilon^{-2}nd\log(n/\delta))$.*

*Proof.* $s = O(\epsilon^{-2}\log(n/\delta))$ is the dimension after transformation. Line 14 is assigning each element of the sketching matrix, and it takes $O(sd) = O(\epsilon^{-2}d\log(n/\delta))$ time since it is a $\mathbb{R}^{s \times d}$ matrix. Line 20 is multiplying the sketching matrix and the vector made up of each coordinate of a node, and it will take $O(sd) = O(\epsilon^{-2}d\log(n/\delta))$. Since we have $n$ nodes to deal with, this whole process will take $O(nsd) = O(\epsilon^{-2}nd\log(n/\delta))$ time. After carefully analyzing the algorithm, it can be deduced that the overall running time complexity is $O(sd + nsd) = O(\epsilon^{-2}nd\log(n/\delta))$. $\qquad\square$

**Lemma 3.8.** *The procedure UPDATE (Algorithm 2) in Theorem 3.6 runs in time $O(\epsilon^{-2}(n+d)\log(n/\delta))$. The total matching value $p$ maintains a $\frac{1-\epsilon}{2}$-approximate matching before calling UPDATE, then $p$ also maintains a $\frac{1-\epsilon}{2}$-approximate matching after calling UPDATE with probability at least $1 - \delta$.*

*Proof.* $s = O(\epsilon^{-2}\log(n/\delta))$ is the dimension after transformation. We call UPDATE when a new node $y$ comes. If $y$ is a buyer node, line 9 is calling QUERY, which takes $O(\epsilon^{-2}(n+d)\log(n/\delta))$ time. Line

---

**Algorithm 1** Initialization Of Fast Greedy

---

1: **data structure** FASTGREEDY
2: **members**
3:     $x_1, x_2, \ldots x_n \in \mathbb{R}^d$ are nodes in the market, and $\widetilde{x}_1, \widetilde{x}_2, \ldots \widetilde{x}_n \in \mathbb{R}^s$ are nodes after sketching.
4:     $m_i$ is the vertex matching with vertex $i$, and $w_i$ is the matching value on $x_i$.
5:     $p$ and $M$ are the matching value and the sketching matrix, respectively.
6:     Let $d_1, d_2, \ldots d_n \in \mathcal{N}$ be the deadline for each node $\quad\quad\triangleright$ Each offline point $x_i$ can only be matched during time $d_i$.
7:     flag[$n$] $\quad\quad\quad\quad\quad\quad\quad\quad\quad\quad\quad\quad\triangleright$ flag[i] decides if node $i$ can be matched.
8: **end members**
9: **procedure** INIT($x_1, \ldots, x_n, \epsilon, \delta$)
10:     $p \leftarrow 0$
11:     $s \leftarrow O(\epsilon^{-2} \log(n/\delta))$
12:     **for** $i = 1, 2, \ldots, s$ **do**
13:         **for** $j = 1, 2, \ldots, d$ **do**
14:             sample $M[i][j]$ from $\{-1/\sqrt{s}, +1/\sqrt{s}\}$ each with $1/2$ probability
15:         **end for**
16:     **end for**
17:     **for** $i = 1, 2, \ldots, n$ **do**
18:         $w_i \leftarrow 0$
19:         flag[$i$] $\leftarrow 1$
20:         $\widetilde{x}_i = M x_i$
21:     **end for**
22: **end procedure**

---

---

**Algorithm 2** Update Of Fast Greedy

---

1: **procedure** UPDATE($y \in \mathbb{R}^d$)
2:     **if** the present time is $d_i$ **then**
3:         flag[$i$] $\leftarrow 0$
4:     **end if**
5:     **if** $y$ is $i$-th seller node **then**
6:         flag[$i$] $\leftarrow 1$
7:     **end if**
8:     **if** $y$ is a buyer node **then**
9:         $\{\widetilde{w}_i\}_{i=1}^n \leftarrow$ QUERY($y$)
10:         $i_0 \leftarrow \arg\max_{\text{flag}[i]=1}\{\widetilde{w}_i - w_i\}$
11:         $m_{i_0} \leftarrow y$
12:         $p \leftarrow p + \max\{w_{i_0}, \widetilde{w}_{i_0}\} - w_{i_0}$
13:         $w_{i_0} \leftarrow \max\{w_{i_0}, \widetilde{w}_{i_0}\}$
14:     **end if**
15: **end procedure**
16: **procedure** QUERY( $y \in \mathbb{R}^d$)
17:     $\widetilde{y} = M y$
18:     **for** $i = 1, 2, \ldots, n$ **do**
19:         **if** flag[$i$] $= 1$ **then** $\widetilde{w}_i \leftarrow \|\widetilde{y} - \widetilde{x}_i\|_2$
20:         **end if**
21:     **end for**
22:     **return** $\{\widetilde{w}_i\}_{i=1}^n$
23: **end procedure**
24: **procedure** TOTALWEIGHT
25:     **return** $p$
26: **end procedure**
27: **end data structure**

---

10 is finding $i_0$ which makes $\widetilde{w}_i - w_i$ maximum if that vertex can still be matched, which takes $O(n)$ time. So, in total, the running time is $O(\epsilon^{-2}(n + d)\log(n/\delta))$.

Then we will prove the second statement. We suppose the total matching value $p$ maintains a $\frac{1-\epsilon}{2}$-approximate matching before calling UPDATE. From Lemma 3.9, we can know that after we call QUERY, we can get $\{\widetilde{w}_i\}_{i=1}^n$ and for $\forall i \in [n]$, $(1-\epsilon)\|y - x_i\|_2 \leq \widetilde{w}_i \leq (1+\epsilon)\|y - x_i\|_2$ with probability $1 - \delta$ at least. According to Lemma 3.1 and Lemma 3.2, the competitive ratio of FASTGREEDY is still $\frac{1-\epsilon}{2}$ after running QUERY. Therefore, the total matching value $p$ still maintains a $\frac{1-\epsilon}{2}$-approximate matching with probability at least $1 - \delta$. □

**Lemma 3.9.** *The procedure QUERY (Algorithm 2) in Theorem 3.6 runs in time $O(\epsilon^{-2}(n + d) \log(n/\delta))$. For $\forall i \in [n]$, it provides estimates estimates $\{\widetilde{w}_i\}_{i=1}^n$ such that: $(1 - \epsilon)\|y - x_i\|_2 \leq \widetilde{w}_i \leq (1 + \epsilon)\|y - x_i\|_2$, with probability at least $1 - \delta$.*

*Proof.* Line 17 is multiplying the sketching matrix with the vector made up of each coordinate of the new buyer node $y$, which takes $O(sd) = O(\epsilon^{-2}d\log(n/\delta))$ time. Line 19 is calculating the weight of edge between the new buyer node $y$ and the seller node still in the market, which takes $O(s) = O(\epsilon^{-2}\log(n/\delta))$ time. Since we need to calculate for $n$ times at most, the whole process will take $O(ns) = O(\epsilon^{-2}n\log(n/\delta))$ time. After carefully analyzing the algorithm, it can be deduced that the overall running time complexity is $O(sd + ns) = O(\epsilon^{-2}(n + d)\log(n/\delta))$.

According to Lemma 2.5, we create a sketching matrix $M \in \mathbb{R}^{s \times d}$, let $f(x) = Mx$ and $w_i = \|y - x_i\|_2$ denote the real matching weight of the edge between seller node $i$ and buyer node $y$, and $\widetilde{w}_i = \|f(y) - f(x_i)\|_2$ denote the approximated weight of the edge between seller node $i$ and buyer node $y$, then there will be $(1 - \epsilon)\|y - x_i\|_2 \leq \widetilde{w}_i \leq (1 + \epsilon)\|y - x_i\|_2$ for $\forall i \in [n]$. Since the failure parameter is $\delta$, for $i \in [n]$ it will provide $(1 - \epsilon)\|y - x_i\|_2 \leq \widetilde{w}_i \leq (1 + \epsilon)\|y - x_i\|_2$, with probability at least $1 - \delta$. □

**Lemma 3.10** (Restatement of Lemma A.5)**.** *The procedure TOTALWEIGHT (Algorithm 2) in Theorem 3.6 runs in time $O(1)$. It outputs a $\frac{1-\epsilon}{2}$-approximate matching with probability at least $1 - \delta$,*

**Lemma 3.11** (Restatement of Lemma A.6, Space storage for FASTGREEDY in Algorithm 1 and Algorithm 2)**.** *The space storage for FASTGREEDY in Algorithm 1 and Algorithm 2 is $O(nd + \epsilon^{-2}(n + d)\log(n/\delta))$.*

**Theorem 3.12.** *Consider the online bipartite matching problem with the inner product weight, for any $\epsilon \in (0, 1)$, $\delta \in (0, 1)$ there exists a data structure FASTPOSTPONEDGREEDY that uses $O(nd + \epsilon^{-2}(n + d)\log(n/\delta))$ space. Assuming there are no offline points at first, each of the online points $x_1, x_2, \ldots, x_n$ will be determined if it is a buyer node or a seller node when it needs to leave the market. FASTPOSTPONEDGREEDY supports the following operations*

- *INIT($\epsilon \in (0, 1), \delta \in (0, 1)$). Given a precision parameter $\epsilon$ and a failure tolerance $\delta$ as inputs, this data structure preprocesses in $O(\epsilon^{-2}d\log(n/\delta))$ time.*

- *UPDATE($x_i \in \mathbb{R}^d$). It takes an online point $x_i$ as inputs and runs in $O(\epsilon^{-2}(n + d)\log(n/\delta))$ time. Let $t \in \mathcal{N}$ denote the arrival time of any online point. QUERY approximately estimates the Euclidean distances from $b_{\widetilde{x}_i}$ to all the data points $s_{\widetilde{x}_1}, s_{\widetilde{x}_2}, \ldots, s_{\widetilde{x}_n} \in R^d$ in time $O(\epsilon^{-2}(n + d)\log(n/\delta))$. For $\forall i \in [n]$, it provides estimates estimates $\{\widetilde{w}_j\}_{j=1}^n$ such that: $(1 - \epsilon)\|s_{x_j} - b_{x_i}\|_2 \leq \widetilde{w}_j \leq (1 + \epsilon)\|s_{x_j} - b_{x_i}\|_2$.*

- *TOTALWEIGHT. It outputs the matching value between offline points and $n$ known online points in $O(1)$ time. It has competitive ratio $\frac{1-\epsilon}{4}$ with probability at least $1 - \delta$.*

To establish the validity of the data structure utilized in Algorithm 3 and its adherence to the conditions specified in Theorem 3.12, we provide a series of lemmas for verification. These lemmas serve as intermediate steps in the overall proof.

**Lemma 3.13.** *The procedure INIT (Algorithm 3) in Theorem 3.12 runs in time $O(\epsilon^{-2}d\log(n/\delta))$.*

*Proof.* $s = O(\epsilon^{-2}\log(n/\delta))$ is the dimension after transformation. Line 15 is assigning each element of the sketching matrix, and it takes $O(sd) = O(\epsilon^{-2}d\log(n/\delta))$ time since it is a $\mathbb{R}^{s \times d}$ matrix. After carefully analyzing the algorithm, it can be deduced that the overall running time complexity is $O(sd) = O(\epsilon^{-2}d\log(n/\delta))$. □

---

**Algorithm 3** Initialization Of Fast Postponed Greedy

---

1:  **data structure** FASTPOSTPONEDGREEDY
2:  **members**
3:      $x_1, x_2, \ldots x_n \in \mathbb{R}^d$                                                    ▷ Nodes in the market
4:      $m_i$ is the vertex matching with vertex $i$, and $w_i$ is the matching value on $x_i$.
5:      $p$ and $M$ are the matching value and the sketching matrix, respectively.
6:      Let $d_1, d_2, \ldots d_n \in \mathcal{N}$ be the deadline for each node          ▷ Each offline point $x_i$ can only be matched during time $d_i$.
7:  **end members**
8:  **procedure** INIT($\epsilon, \delta$)
9:      $p \leftarrow 0$
10:     $s \leftarrow O(\epsilon^{-2} \log(n/\delta))$
11:     $\mathcal{S}_0 \leftarrow \emptyset$.
12:     $\mathcal{B}_0 \leftarrow \emptyset$.
13:     **for** $i = 1, 2, \ldots, s$ **do**
14:         **for** $j = 1, 2, \ldots, d$ **do**
15:             sample $M[i][j]$ from $\{-1/\sqrt{s}, +1/\sqrt{s}\}$ each with $1/2$ probability
16:         **end for**
17:     **end for**
18: **end procedure**

---

**Lemma 3.14** (Restatement of Lemma A.7)**.** *The procedure* UPDATE *(Algorithm 3) in Theorem 3.12 runs in time $O(\epsilon^{-2}(n + d) \log(n/\delta))$. The total matching value $p$ maintains a $\frac{1-\epsilon}{4}$-approximate matching before calling* UPDATE*, then $p$ also maintains a $\frac{1-\epsilon}{4}$-approximate matching after calling* UPDATE *with probability at least $1 - \delta$.*

**Lemma 3.15** (Restatement of Lemma A.8)**.** *The procedure* QUERY *(Algorithm 4) in Theorem 3.12 runs in time $O(\epsilon^{-2} n \log(n/\delta))$. For $\forall j \in [n]$, it provides estimates $\{\widetilde{w}_j\}_{j=1}^n$ such that: $(1 - \epsilon)\|s_{x_j} - b_{x_i}\|_2 \leq \widetilde{w}_j \leq (1 + \epsilon)\|s_{x_j} - b_{x_i}\|_2$, with probability at least $1 - \delta$.*

**Lemma 3.16** (Restatement of Lemma A.9)**.** *The procedure* TOTALWEIGHT *(Algorithm 4) in Theorem 3.12 runs $O(1)$ time. It outputs a $\frac{1-\epsilon}{2}$-approximate matching with probability at least $1 - \delta$.*

**Lemma 3.17** (Restatement of Lemma A.10, Space storage for FASTPOSTPONEDGREEDY in Algorithm 3 and Algorithm 4)**.** *The space storage for FASTPOSTPONEDGREEDY is $O(nd + \epsilon^{-2}(n + d) \log(n/\delta))$.*

# 4. Experiments

After presenting these theoretical analyses, we now move on to the experiments.

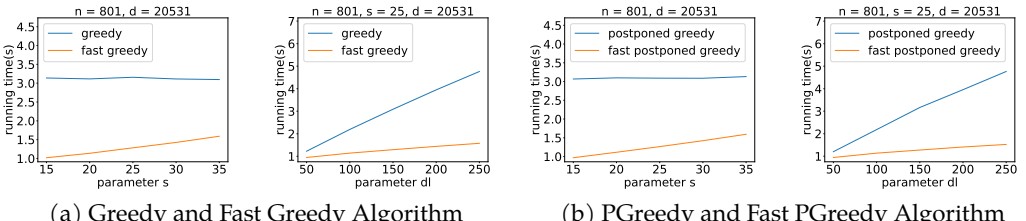

(a) Greedy and Fast Greedy Algorithm              (b) PGreedy and Fast PGreedy Algorithm

Figure 1: The relationship between running time and parameter $s$ and $dl$ on GECRS data set. The parameters are defined as follows: $n$ is the node count, $d$ is the original node dimension, $s$ is the dimension after transformation, and $dl$ is the maximum matching time per node (referred to as the deadline). Here GECRS denotes gene expression cancer RNA-Seq Data Set. PGreedy denotes Postponed Greedy.

**Purpose.** This section furnishes a systematic evaluation of our optimized algorithms' performance using real-world data sets. We employ a sketching matrix to curtail the vector entries, thus expediting

**Algorithm 4** Update Of Fast Postponed Greedy

1: **procedure** UPDATE($x_i \in \mathbb{R}^d$)
2:     Set the status of $x_i$ to be undetermined.
3:     $\widetilde{x}_i \leftarrow M x_i$
4:     $\mathcal{S}_t \leftarrow \mathcal{S}_{t-1} \cup s_{\widetilde{x}_i}$                                             ▷ Add a seller copy.
5:     $m_{s_{\widetilde{x}_i}} \leftarrow$ null
6:     $w_i \leftarrow 0$
7:     $\mathcal{B}_t \leftarrow \mathcal{B}_{t-1} \cup b_{\widetilde{x}_i}$                                             ▷ Add a buyer copy
8:     **if** the present time is $d_i$ **then**
9:         **if** status of point $x_i$ is undetermined **then**
10:             set it to be either seller or buyer with probability $1/2$ each.
11:         **end if**
12:         **if** $m_{\widetilde{x}_i} \neq$ null **then**
13:             $b_{\widetilde{x}_l} \leftarrow m_{\widetilde{x}_i}$.
14:             **if** $x_i$ is a seller **then**
15:                 $p \leftarrow p + \widetilde{w}_i$
16:                 Finalize the matching of point $x_i$ with point $x_l$. Set the status of point $x_l$ to be a buyer.
17:             **end if**
18:             **if** $x_i$ is a buyer **then**
19:                 Set the status of point $x_l$ to be a seller.
20:             **end if**
21:         **end if**
22:         $\mathcal{S}_t \leftarrow \mathcal{S}_t \backslash \{s_{\widetilde{x}_i}\}$                                 ▷ Remove the seller and buyer copies.
23:         $\mathcal{B}_t \leftarrow \mathcal{B}_t \backslash \{b_{\widetilde{x}_i}\}$
24:     **end if**
25:     $\{\widetilde{w}_j\}_{j=1}^n \leftarrow$ QUERY($b_{\widetilde{x}_i}$)
26:     $j_0 \leftarrow \arg \max_{s_{\widetilde{x}_j} \in \mathcal{S}_t} \{\widetilde{w}_j - w_j\}$
27:     $m_{\widetilde{x}_{j_0}} \leftarrow b_{\widetilde{x}_i}$
28:     $w_{j_0} \leftarrow \max\{w_{j_0}, \widetilde{w}_{j_0}\}$
29: **end procedure**
30: **procedure** QUERY( $b_{\widetilde{x}_i} \in \mathbb{R}^s$)
31:     **for** $j = 1, 2, \ldots, n$ **do**
32:         **if** $s_{\widetilde{x}_j} \in \mathcal{S}_t$ **then**
33:             $\widetilde{w}_j \leftarrow \|b_{\widetilde{x}_i} - s_{\widetilde{x}_j}\|_2$
34:         **end if**
35:     **end for**
36:     **return** $\{\widetilde{w}_j\}_{j=1}^n$
37: **end procedure**
38: **procedure** TOTALWEIGHT
39:     **return** $p$
40: **end procedure**
41: **end data structure**

Table 1: The parameters are defined as follows: $n$ is the node count, $d$ is the original node dimension, $s$ is the dimension after transformation, and dl is the maximum matching time per node (referred to as the deadline). Here GECRS denotes gene expression cancer RNA-Seq Data Set. Let ARBT denote a study of Asian Religious and Biblical Texts Data Set.

| Dataset Names | $n$ | $d$ | $s$ | dl |
|---|---|---|---|---|
| GECRS | 801 | 20351 | $[15, 20, 25, 30, 35]$ | $[50, 100, 150, 200, 250]$ |
| Arcene | 700 | 10000 | $[20, 30, 40, 50, 60]$ | $[200, 300, 400, 500, 600]$ |
| ARBT | 590 | 8265 | $[10, 20, 30, 40, 50]$ | $[50, 100, 150, 200, 250]$ |
| REJAFADA | 1996 | 6826 | $[20, 30, 40, 50, 60]$ | $[50, 100, 150, 200, 250]$ |

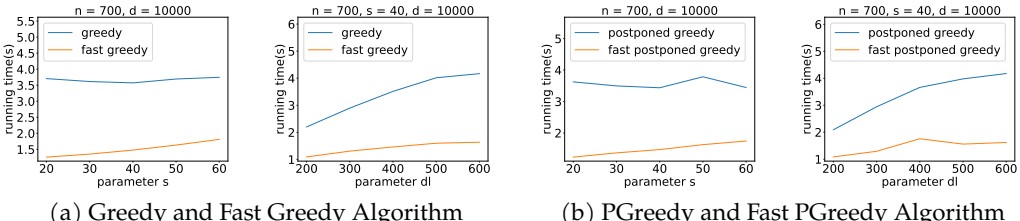

(a) Greedy and Fast Greedy Algorithm        (b) PGreedy and Fast PGreedy Algorithm

Figure 2: The relationship between running time and parameter $s$ and $dl$ on Arcene data set. The parameters are defined as follows: $n$ is the node count, $d$ is the original node dimension, $s$ is the dimension after transformation, and $dl$ is the maximum matching time per node (referred to as the deadline).

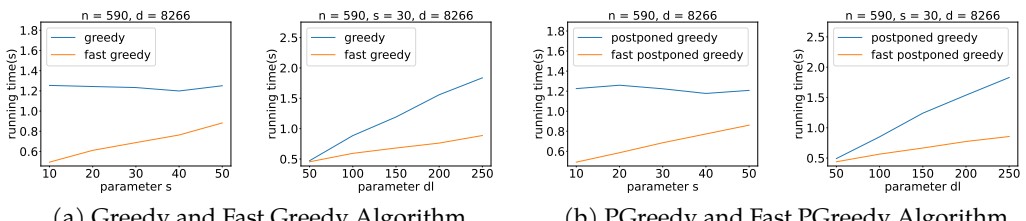

(a) Greedy and Fast Greedy Algorithm        (b) PGreedy and Fast PGreedy Algorithm

Figure 3: The relationship between running time and parameter $s$ and $dl$ on ARBT data set. The parameters are defined as follows: $n$ is the node count, $d$ is the original node dimension, $s$ is the dimension after transformation, and $dl$ is the maximum matching time per node (referred to as the deadline). Let ARBT denote a study of Asian Religious and Biblical Texts Data Set.

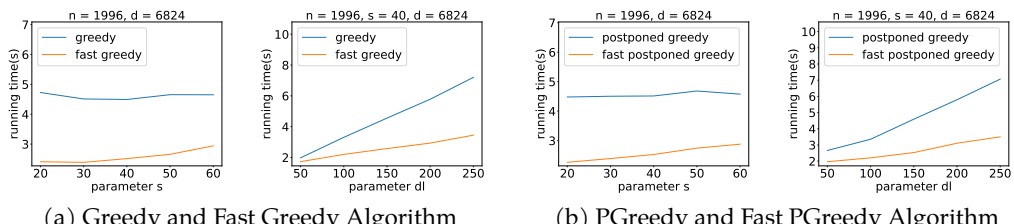

(a) Greedy and Fast Greedy Algorithm        (b) PGreedy and Fast PGreedy Algorithm

Figure 4: The relationship between running time and parameter $s$ and $dl$ on REJAFADA data set. The parameters are defined as follows: $n$ is the node count, $d$ is the original node dimension, $s$ is the dimension after transformation, and $dl$ is the maximum matching time per node (referred to as the deadline).

the resolution of online weighted bipartite problems amidst substantial data quantities. Detailed accounts of the synthetic data set experimentation are provided in the appendix. Here, we encapsulate the results obtained from real-world data sets:

- For small enough $s$, our algorithms demonstrate superior speed compared to their original counterparts.

- All four algorithms exhibit linear escalation in running time with an increase in $n$.

- For our algorithms, the running time escalates linearly with $s$, an effect absent in the original algorithms.

- The original algorithms exhibit a linear increase in running time with $d$, and our algorithms follow suit when $n$ and $s$ are substantially smaller compared to $d$.

- Denoting the parameter deadline as dl, it is observed that the running time of all four algorithms intensifies with dl.

**Configuration.** Our computations utilize an apparatus with an AMD Ryzen 7 4800H CPU and an RTX 2060 GPU, implemented on a laptop. The device operates on the Windows 11 Pro system, with Python serving as the programming language. We represent the distance weights as $\ell_2$. The symbol $n$ signifies the count of vectors in both partitions of the bipartite graph, thereby implying an equal quantity of buyers and sellers.

**Real data sets.** In this part, we run all four algorithms on real data sets from the UCI library [46] to observe if our algorithms are better than the original ones in a real-world setting.

- Gene expression cancer RNA-Seq Data Set [47]: This data set is structured with samples stored in a row-wise manner. Each sample's attributes are its RNA-Seq gene expression levels, measured via the Illumina HiSeq platform.
- Arcene Data Set [48]: The Arcene data set is derived from three merged mass-spectrometry data sets, ensuring a sufficient quantity of training and testing data for a benchmark. The original features represent the abundance of proteins within human sera corresponding to a given mass value. These features are used to distinguish cancer patients from healthy individuals. Many distractor features, referred to as 'probes' and devoid of predictive power, have been included. Both features and patterns have been randomized.
- A study of Asian Religious and Biblical Texts Data Set [49]: This data set encompasses $580$ instances, each with $8265$ attributes. The majority of the sacred texts in the data set were sourced from Project Gutenberg.
- REJAFADA Data Set [50]: The REJAFADA (Retrieval of Jar Files Applied to Dynamic Analysis) is a data set designed for the classification of Jar files as either benign or malware. It consists of $998$ malware Jar files and an equal number of benign Jar files.

**Results for Real Data Set.** We run GREEDY, FASTGREEDY, POSTPONEDGREEDY (PGreedy for shorthand in Figure 1, 2, 3, 4.) and FASTPOSTPONEDGREEDY (Fast PGreedy for shorthand in Figure 1, 2, 3, 4.) algorithms on GECRS, Arcene, ARBT, and REJAFADA data sets respectively. Specifically, Fig. 1a and Fig. 1b illustrate the relationship between the running time of the four algorithms and the parameters $s$ and dl on the GECRS data set, respectively. Similarly, Fig. 2a and Fig. 2b showcase the running time characteristics of the algorithms with respect to the parameters $s$ and dl on the Arcene data set. Analogously, Fig. 3a and Fig. 3b represent the relationship between the running time and parameters $s$ and dl on the ARBT data set. Lastly, Fig. 4a and Fig. 4b exhibit the running time variations concerning the parameters $s$ and dl on the REJAFADA data set. The results consistently indicate that our proposed FASTGREEDY and FASTPOSTPONEDGREEDY algorithms exhibit significantly faster performance compared to the original GREEDY and POSTPONEDGREEDY algorithms when applied to real data sets. These findings highlight the superiority of our proposed algorithms in terms of efficiency when dealing with real-world data scenarios.

## 5. Conclusion

In this paper, we study the online matching problem with *deadline* and solve it with a sketching matrix. We provided a new way to compute the weight of the edge between two nodes in a bipartite. Compared with original algorithms, our algorithms optimize the time complexity from $O(nd)$ to $\widetilde{O}(\epsilon^{-2} \cdot (n+d))$. Furthermore, the total weight of our algorithms is very close to that of the original algorithms respectively, which means the error caused by the sketching matrix is very small. Our algorithms can also be used in areas like recommending short videos to users.

For matching nodes with many entries, the experiment result shows that our algorithms only take us a little time if parameter $s$ is small enough. We think generalizing our techniques to other variations of the matching problem could be an interesting future direction. We remark that implementing our algorithm would have carbon release to the environment.

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

# Appendix

**Roadmap.** In Section A, we also provide more proof, such as Theorem 3.5, Lemma 3.11 and Lemma 3.17 which prove the correctness of FASTPOSTPONEDGREEDY, the space storage of FASTGREEDY, and the space storage of FASTPOSTPONEDGREEDY respectively.

In Section B, to illustrate the impact of various parameters, namely $n$, $s$, $d$, and $\mathrm{dl}$, we present additional experimental results. The parameters are defined as follows: $n$ is the node count, $d$ is the original node dimension, $s$ is the dimension after transformation, and $\mathrm{dl}$ is the maximum matching time per node (referred to as the deadline).

## A. Missing Proofs

In Section A.1, we provide the proof for Lemma 3.1. In Section A.2, we provide the proof for Lemma 3.2. In Section A.3, we provide the proof for Lemma 3.3. In Section A.4, we provide the proof for Theorem 3.5. In Section A.5, we provide the proof for Lemma 3.10. In Section A.6, we provide the proof for Lemma 3.11. In Section A.7, we provide the proof for Lemma 3.14. In Section A.8, we provide the proof for Lemma 3.15. In Section A.9, we provide the proof for Lemma 3.16. In Section A.10, we provide the proof for Lemma 3.17.

### A.1. Proof of Lemma 3.1

**Lemma A.1** (Restatement of Lemma 3.1). *STANDARDGREEDY in Algorithm 5 is a $1/2$-competitive algorithm.*

*Proof.* We use $w_s$ to denote the current matching value of seller $s$. Let $w_{s,b}$ denote the weight of the edge between the buyer node and $s$-th seller node. Let $i(b)$ be the increment of matching value when buyer $b$ arrives: $i(b) = \max_{s \in \mathcal{S}}\{w_{s,b}, w_s\} - w_s$. Let $v_f(s)$ be the final matching value for a seller node $s$. If we add all the increments of matching value that every buyer node brings, we will get the sum of every seller node's matching value. $\sum_{b \in \mathcal{B}} i(b) = \sum_{s \in \mathcal{S}} v_f(s)$. Let alg be the weight of the matching in STANDARDGREEDY. Let $\mathrm{alg} = \sum_{s \in \mathcal{S}} v_f(s)$. Let opt be the optimal matching weight of any matching that can be constructed. Let $\lambda_i$ denote the arrival time of node $i$. We need to do the following things to acquire the maximum of opt:

$$
\begin{aligned}
\text{minimize} \quad & \sum_{i \in [n]} a_i \\
\text{subject to } & a_i \geq 0 & \forall i \in [n] \\
& |\lambda_i - \lambda_j| \leq \mathrm{dl}, & \forall i < j \\
& a_i + a_j \geq w_{i,j} & \forall i, j \in [n].
\end{aligned}
$$

During the whole process, we update the value of $w_s$ only when $w_s \leq w_{s,b}$. So $v_f(s) \geq w_s$. Since $i(b)$ is the maximum of $w_{s,b} - w_s$, we can conclude that $i(b) \geq w_{s,b} - w_s$. Therefore, $v_f(s) + i(b) \geq w_{s,b}$ and $\{v_f(s)\}_{s \in \mathcal{S}} \cup \{i(b)\}_{b \in \mathcal{B}}$ is a feasible solution.

We conclude

$$
\mathrm{opt} \leq \sum_{b \in \mathcal{B}} i(b) + \sum_{s \in \mathcal{S}} v_f(s) = 2 \sum_{s \in \mathcal{S}} v_f(s) = 2\mathrm{alg},
$$

where the first step follows from $v_f(s) + i(b) \geq w_{s,b}$, the second step follows from $\sum_{s \in \mathcal{S}} v_f(s) = \sum_{b \in \mathcal{B}} i(b)$, and the last step follows from $\mathrm{alg} = \sum_{s \in \mathcal{S}} v_f(s)$. □

### A.2. Proof of Lemma 3.2

**Lemma A.2** (Restatement of Lemma 3.2). *Let $w_{i,j}$ denote the real weight of the edge between seller $i$ and buyer $j$, and $\widetilde{w}_{i,j}$ denote the approximated weight. Let $\epsilon \in (0, 0.1)$ denote the precision parameter. If for all*

$\epsilon$, *there exists an $\alpha$-approximation algorithm for the online weighted matching problem and a $\delta > 0$, where $(1 - \delta)w_{i,j} \leq \widetilde{w}_{i,j} \leq (1 + \delta)w_{i,j}$ for any seller node $i$ and buyer node $j$, there exists a greedy algorithm with competitive ratio $\alpha(1 - \epsilon)$.*

*Proof.* Since $\epsilon \in (0, 0.1)$ and we have $(1 - \delta)w_{i,j} \leq \widetilde{w}_{i,j} \leq (1 + \delta)w_{i,j}$. Let alg and opt denote the matching weight and optimal weight of the original problem $G = (\mathcal{S}, \mathcal{B}, E)$ respectively. Then we define alg$'$ is the matching weight of another problem $G' = (\mathcal{S}, \mathcal{B}, E')$, and our algorithm gives the optimal weight opt$'$. Since for each edge $\widetilde{w}_{i,j} \leq (1 + \delta)w_{i,j}$ and alg $= \sum_{\substack{i \in \mathcal{S} \\ j \in \mathcal{B}}} w_{i,j}$, we can know that alg$' \leq (1 + \delta)$alg. Since for each edge $(1 - \delta)w_{i,j} \leq \widetilde{w}_{i,j}$ and opt $= \sum_{\substack{i \in \mathcal{S} \\ j \in \mathcal{B}}} w_{i,j}$, we can know that opt$' \geq (1 - \delta)$opt. Thus,

$$\text{alg} \geq \frac{1}{1 + \delta}\text{alg}' \geq \frac{\alpha}{1 + \delta}\text{opt}' \geq \frac{\alpha(1 - \delta)}{1 + \delta}\text{opt} \geq \alpha(1 - 2\delta)\text{opt},$$

where the first step follows from alg$' \leq (1 + \delta)$alg, the second step follows from there exists a $\alpha$-approximation greedy algorithm, the third step follows from opt$' \geq (1 - \delta)$opt, and the last step follows from $\frac{\alpha(1-\delta)}{1+\delta} \geq \frac{\alpha(1-\delta)^2}{1-\delta^2} \geq \frac{\alpha(1-\delta)^2}{2} \geq \alpha(1 - 2\delta)$. Then we can draw our conclusion because we can replace $2\delta$ with a new $\epsilon = 2\delta$ in the last step. $\square$

## A.3. Proof of Lemma 3.3

**Lemma A.3** (Restatement of Lemma 3.3). *If a node is determined to be a seller or a buyer with $1/2$ probability in STANDARDGREEDY in Algorithm 5, then this new STANDARDPOSTPONEDGREEDY algorithm is a $1/4$-competitive algorithm.*

*Proof.* Let fpg denote the weight constructed by FASTPOSTPONEDGREEDY algorithm. Let alg and opt denote the matching weight and optimal weight of original problem $G = (\mathcal{S}, \mathcal{B}, E)$ respectively.

Since the probability of a node being a seller is $1/2$, we can know that

$$\text{fpg} = \mathbb{E}[\sum_{i \in \mathcal{S}} w_{i,j}] = \frac{1}{2} \sum_{i \in \mathcal{S}, j \in \mathcal{B}} w_{i,j} = \frac{1}{2}\text{alg} \geq \frac{1}{4}\text{opt}$$

where the first step follows from the definition of fpg, the second step follows from the probability of a node being a seller is $1/2$, the third step follows from alg $= \sum_{i \in \mathcal{S}, j \in \mathcal{B}} w_{i,j}$, and the last step follows from Lemma 3.1. $\square$

## A.4. Proof of Theorem 3.5

**Theorem A.4** (Restatement of Theorem 3.5, Correctness of FASTPOSTPONEDGREEDY in Algorithm 3 and Algorithm 4). *FASTPOSTPONEDGREEDY is a $\frac{1-\epsilon}{4}$-competitive algorithm.*

*Proof.* According to Lemma 3.3, there exists a $\frac{1}{4}$-competitive algorithm. According to Lemma 2.5, we create a sketching matrix $M \in \mathbb{R}^{s \times d}$, let $f(x) = Mx$ and $w_i = \|y_j - x_i\|_2$ denote the real matching weight of the edge between seller node $x_i$ and buyer node $y_j$, and $\widetilde{w}_{i,j} = \|f(y_j) - f(x_i)\|_2$ denote the approximated weight of the edge between seller node $x_i$ and buyer node $y_j$, then there will be $(1 - \epsilon)w_{i,j} \leq \widetilde{w}_{i,j} \leq (1 + \epsilon)w_{i,j}$ for $\forall i, j \in [n]$. Based on the implications of Lemma 3.2, we can confidently assert that the competitive ratio of FASTPOSTPONEDGREEDY is $\frac{1-\epsilon}{4}$. $\square$

## A.5. Proof of Lemma 3.10

**Lemma A.5** (Restatement of Lemma 3.10). *The procedure TOTALWEIGHT (Algorithm 2) in Theorem 3.6 runs in time $O(1)$. It outputs a $\frac{1-\epsilon}{2}$-approximate matching with probability at least $1 - \delta$,*

*Proof.* Line 25 is returning the total matching value $p$ which has already been calculated after calling UPDATE, which requires $O(1)$ time.

The analysis demonstrates that the overall running time complexity is constant, denoted as $O(1)$.

According to Lemma 3.8, we can know that the total weight $p$ maintains a $\frac{1-\epsilon}{2}$-approximate matching after calling UPDATE. So when we call TOTALWEIGHT, it will output a $\frac{1-\epsilon}{2}$-approximate matching if it succeeds.

The probability of successful running of UPDATE is at least $1 - \delta$, so the probability of success is $1 - \delta$ at least. $\qquad\square$

## A.6. Proof of Lemma 3.11

**Lemma A.6** (Restatement of Lemma 3.11, Space storage for FASTGREEDY in Algorithm 1 and Algorithm 2 ). *The space storage for FASTGREEDY in Algorithm 1 and Algorithm 2 is $O(nd + \epsilon^{-2}D^2(n + d)\log(n/\delta))$.*

*Proof.* $s = O(\epsilon^{-2}\log(n/\delta))$ is the dimension after transformation. In FASTGREEDY, we need to store a $\mathbb{R}^{s \times d}$ sketching matrix, multiple arrays with $n$ elements, two point sets containing $n$ nodes with $d$ dimensions, and a point set containing $n$ nodes with $s$ dimensions. The total storage is

$$O(n + sd + nd + ns) = O(nd + \epsilon^{-2}(n + d)\log(n/\delta)).$$

$\qquad\square$

## A.7. Proof of Lemma 3.14

**Lemma A.7** (Restatement of Lemma 3.14). *The procedure UPDATE (Algorithm 3) in Theorem 3.12 runs in time $O(\epsilon^{-2}(n + d)\log(n/\delta))$. The total matching value $p$ maintains a $\frac{1-\epsilon}{4}$-approximate matching before calling UPDATE, then $p$ also maintains a $\frac{1-\epsilon}{4}$-approximate matching after calling UPDATE with probability at least $1 - \delta$.*

*Proof.* $s = O(\epsilon^{-2}\log(n/\delta))$ is the dimension after transformation. We call UPDATE when a new node $x_i$ comes. Line 3 is multiplying the sketching matrix and the vector made up of each coordinate of a node, and it will take $O(sd) = O(\epsilon^{-2}d\log(n/\delta))$ time. Line 25 is calling QUERY, which requires $O(\epsilon^{-2}n\log(n/\delta))$ time. Line 26 is finding $j_0$ which makes $\widetilde{w}_j - w_j$ maximum if that vertex can still be matched, which requires $O(n)$ time.

So, in total, the running time is

$$O(sd + ns + n) = O(\epsilon^{-2}(n + d)\log(n/\delta)).$$

Then we will prove the second statement.

We suppose the total matching value $p$ maintains a $\frac{1-\epsilon}{4}$-approximate matching before calling UPDATE. From Lemma 3.15, we can know that after we call QUERY, we can get $\{\widetilde{w}_j\}_{j=1}^n$ and for $\forall j \in [n](1 - \epsilon)\|s_{x_j} - b_{x_i}\|_2 \leq \widetilde{w}_j \leq (1 + \epsilon)\|s_{x_j} - b_{x_i}\|_2$ with probability $1 - \delta$ at least. According to Lemma 3.3 and Lemma 3.2, the competitive ratio of FASTPOSTPONEDGREEDY is still $\frac{1-\epsilon}{4}$ after running QUERY. Therefore, the total matching value $p$ still maintains a $\frac{1-\epsilon}{4}$-approximate matching with probability at least $1 - \delta$. $\qquad\square$

## A.8. Proof of Lemma 3.15

**Lemma A.8** (Restatement of Lemma 3.15). *The procedure QUERY (Algorithm 4) in Theorem 3.12 runs in time $O(\epsilon^{-2}n\log(n/\delta))$. For $\forall j \in [n]$, it provides estimates estimates $\{\widetilde{w}_j\}_{j=1}^n$ such that:*

$$(1 - \epsilon)\|s_{x_j} - b_{x_i}\|_2 \leq \widetilde{w}_j \leq (1 + \epsilon)\|s_{x_j} - b_{x_i}\|_2$$

*with probability at least $1 - \delta$.*

*Proof.* Line 33 is calculating the weight of the edge between the buyer node copy $b_{x_i}$ and the seller node still in the market, which requires $O(s) = O(\epsilon^{-2}\log(n/\delta))$ time. Since we need to calculate for $n$ times at most, the whole process will take $O(ns) = O(\epsilon^{-2}n\log(n/\delta))$ time.

So, in total, the running time is $O(ns) = O(\epsilon^{-2}(n+d)\log(n/\delta))$.

According to Lemma 2.5, we create a sketching matrix $M \in \mathbb{R}^{s \times d}$, let $f(x) = Mx$ and $w_j = \|s_{x_j} - b_{x_i}\|_2$ denote the real matching value of seller node $s_{x_j}$ when buyer node $b_{x_i}$ comes, and $\widetilde{w}_j = \|f(b_{x_i}) - f(s_{x_j})\|_2$ denote the approximated matching value of seller node $s_{x_j}$ when buyer node $b_{x_i}$ comes, then there will be $(1 - \epsilon)\|s_{x_j} - b_{x_i}\|_2 \le \widetilde{w}_j \le (1 + \epsilon)\|s_{x_j} - b_{x_i}\|_2$ for $\forall i \in [n]$.

Since the failure parameter is $\delta$, for $j \in [n]$ it will provide

$$(1 - \epsilon)\|s_{x_j} - b_{x_i}\|_2 \le \widetilde{w}_j \le (1 + \epsilon)\|s_{x_j} - b_{x_i}\|_2$$

with probability at least $1 - \delta$. $\qquad\square$

### A.9. Proof of Lemma 3.16

**Lemma A.9** (Restatement of Lemma 3.16). *The procedure* TOTALWEIGHT (*in Algorithm 4*) *in Theorem 3.12 runs in time $O(1)$. It outputs a $\frac{1-\epsilon}{2}$-approximate matching with probability at least $1 - \delta$,*

*Proof.* Line 39 is returning the total matching value $p$ which has already been calculated after calling UPDATE, which requires $O(1)$ time.

The analysis demonstrates that the overall running time complexity is constant, denoted as $O(1)$.

According to Lemma 3.14, we can know that the total weight $p$ maintains a $\frac{1-\epsilon}{2}$-approximate matching after calling UPDATE. So when we call TOTALWEIGHT, it will output a $\frac{1-\epsilon}{4}$-approximate matching if it succeeds.

The probability of successful running of UPDATE is at least $1 - \delta$, so the probability of success is $1 - \delta$ at least. $\qquad\square$

### A.10. Proof of Lemma 3.17

**Lemma A.10** (Restatement of Lemma 3.17, Space storage for FASTPOSTPONEDGREEDY in Algorithm 3 and Algorithm 4). *The space storage for* FASTPOSTPONEDGREEDY *is $O(nd + \epsilon^{-2}D^2(n+d)\log(n/\delta))$.*

*Proof.* $s = O(\epsilon^{-2}D^2\log(n/\delta))$ is the dimension after transformation. In FASTPOSTPONEDGREEDY, we need to store a $\mathbb{R}^{s \times d}$ sketching matrix, multiple arrays with $n$ elements, a point set containing $n$ nodes with $d$ dimensions, and two point sets containing $n$ nodes with $s$ dimensions. The total storage is

$$O(n + sd + nd + ns) = O(nd + \epsilon^{-2}D^2(n+d)\log(n/\delta)).$$

$\qquad\square$

## B. More Experiments

**Data Generation.** At the beginning, we generate $n$ random vectors of $x_1, \cdots, x_n \in \mathbb{R}^d$ for seller set and $y_1, \cdots, y_n \in \mathbb{R}^d$ for buyer set. Each vector follows the subsequent procedure:

- Each coordinate of the vector is selected uniformly from the interval $[-1, 1]$.
- Normalization is applied to each vector such that its $\ell_2$ norm becomes 1.

---

**Algorithm 5** Standard Greedy Algorithm

---

1: **data structure** STANDARDGREEDY
2: **members**
3:     $x_1, x_2, \ldots x_n \in \mathbb{R}^d$                                                    ▷ Nodes in the market
4:     $w_i$                                                                  ▷ Matching value on $x_i$ seller node
5:     $m(i)$                                                              ▷ The vertex matching with vertex $i$
6:     $p$                                                                           ▷ Total matching value
7:     Let $d_1, d_2, \ldots d_n \in \mathcal{N}$ be the deadline for each node      ▷ Each offline point $x_i$ can only be
     matched during time $d_i$.
8:     flag[$n$]                                                        ▷ flag[i] decides if node $i$ can be matched.
9: **end members**
10: **procedure** INIT($x_1, \ldots, x_n,$)
11:     **for** $i = 1, 2, \ldots, n$ **do**
12:         $w_i \leftarrow 0$
13:         flag[$i$] $\leftarrow 1$
14:     **end for**
15:     $p \leftarrow 0$
16: **end procedure**
17: **procedure** UPDATE($b \in \mathbb{R}^d$)
18:     **if** the present time is $d_i$ **then**
19:         flag[$i$] $\leftarrow 0$
20:     **end if**
21:     $i_0 \leftarrow \arg\max_{\text{flag}[i]=1}\{w_{i,b} - w_i\}$
22:     $m_{i_0} \leftarrow b$
23:     $p \leftarrow p + \max\{w_{i_0}, w_{i_0,b}\} - w_{i_0}$
24:     $w_{i_0} \leftarrow \max\{w_{i_0}, w_{i_0,b}\}$
25: **end procedure**
26: **procedure** QUERY( )
27:     **return** $p$
28: **end procedure**
29: **end data structure**

---

Given that the vectors are randomly generated, the order of their arrival does not affect the experimental results. We define that $i$-th vector of a set in each set comes at time $i$.

In the previous algorithm, for $i$-th vector of the seller set and $j$-th vector of the buyer set, we define the weights between the two nodes to be $\|x_i - y_j\|_2$, which requires $O(nd)$ time.

In the new algorithm, we choose a sketching matrix $S \in \mathbb{R}^{s \times d}$ whose entry is sampled from $\{-1/\sqrt{s}, +1/\sqrt{s}\}$. We have already known $x_1, \cdots, x_n$ at the beginning, and we precompute $\widetilde{x}_i = Sx_i$ for $\forall i \in [n]$. In each iteration, when a buyer $y_j$ comes, we compute $\widetilde{y}_j = Sy_j$ and use $\|\widetilde{x}_i - \widetilde{y}_j\|_2$ to estimate the weight. This takes $O((n + d)s)$ time. After we calculate the weight, we need to find the seller to make the weight between it and the current buyer maximum.

**Parameter Configuration.** In our experimental setup, we select the following parameter values: $n = 1000$, $d = 50000$, and $s = 20$ as primary conditions.

**Results for Synthetic Data Set.** To assess the performance of our algorithms, we conducted experiments comparing their running time with that of the original algorithms. We decide that the $i$-th node of a set comes at $i$-th iteration during our test. Fig. 5a and Fig. 5b illustrate that our algorithms consistently outperformed the original algorithms in terms of running time at each iteration. Specifically, our algorithms exhibited running times equivalent to only $10.0\%$ and $6.0\%$ of the original algorithms, respectively.

Additionally, we analyzed the running time of each algorithm per iteration. Fig. 5a and Fig. 5b depict the running time of each algorithm during per iteration. The results demonstrate that our algorithms consistently outperformed the original algorithms in terms of running time per iteration. dl the maximum matching time per node (referred to as the deadline). The dl is randomly chosen from

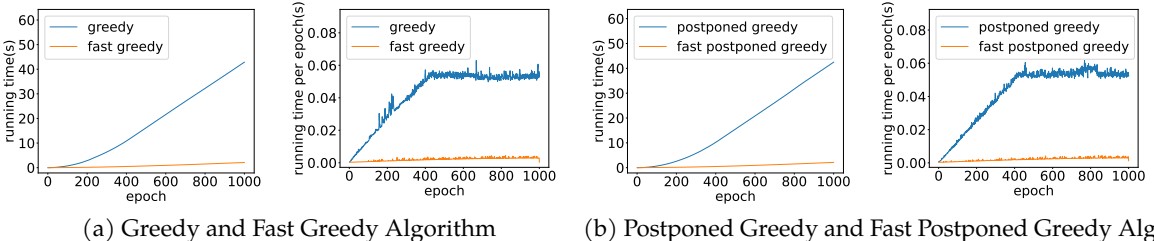

(a) Greedy and Fast Greedy Algorithm

(b) Postponed Greedy and Fast Postponed Greedy Algorithm

Figure 5: Comparison between the running time of each two algorithms.

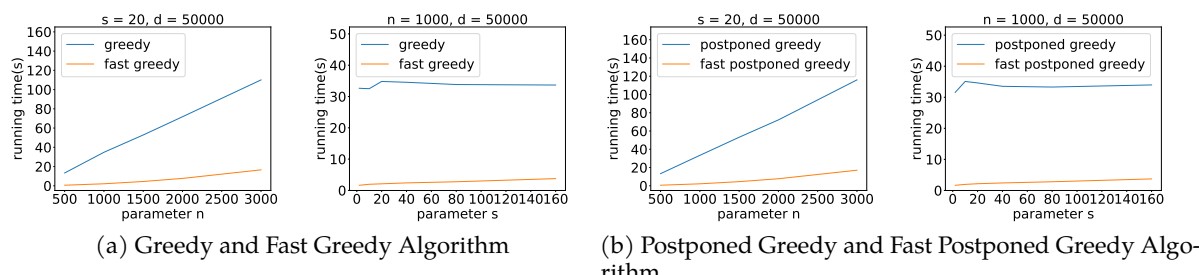

(a) Greedy and Fast Greedy Algorithm

(b) Postponed Greedy and Fast Postponed Greedy Algorithm

Figure 6: The relationship between running time and parameter $n$ and $s$. $n$ is the node count, and $s$ is the dimension after transformation.

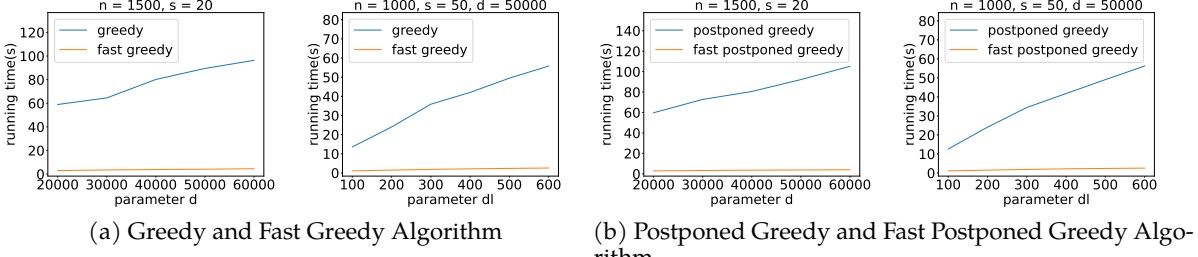

(a) Greedy and Fast Greedy Algorithm

(b) Postponed Greedy and Fast Postponed Greedy Algorithm

Figure 7: The relationship between running time and parameter $d$ and $\mathrm{dl}$. $d$ is the original node dimension, and $\mathrm{dl}$ is the maximum matching time per node (referred to as the deadline).

$[1, n]$. In this case, the $\mathrm{dl}$ of original algorithms is $420$. The running time of per iteration increases as the number of iterations increases until it reaches $\mathrm{dl}$. This is because all nodes are required to remain in the market for $\mathrm{dl}$ iterations and are not matched thereafter.

Furthermore, we investigated the influence of the parameter $n$ while keeping the other parameters constant. Fig. 6a and Fig. 6b illustrate that our algorithms consistently outperformed the original algorithms when $n \in [500, 3000]$. Additionally, the running time of each algorithm exhibited a linear increase with the growth of $n$, which aligns with our initial expectations.

We also examined the impact of the parameter $s$ by varying its value while keeping the other parameters constant. Fig. 6a and Fig. 6b reveal that $s$ does not significantly affect the running time of the original algorithms but does impact the running time of our algorithms. Notably, our algorithms demonstrate faster performance when $s$ is smaller.

To evaluate the influence of the parameter $d$, we varied its value while running all four algorithms. Throughout our tests, we observed that the impact of $d$ was considerably less pronounced compared

to $n$ and $s$. Consequently, we set $d$ to a sufficiently large value relative to $n$ and $s$. As shown in Fig. 7a and Fig. 7b, indicate that $d$ does not significantly affect the running time of our algorithms, but it does have a considerable influence on the running time of the original algorithms.

We conducted experiments to assess the influence of the parameter $dl$ while keeping the other parameters constant. $dl$ was selected from the interval $[1, n]$ as choosing values outside this range would render the parameter irrelevant since all nodes would remain in the market throughout the entire process. Fig. 7a and Fig. 7b show that our algorithms are faster than original algorithms when $dl \in [200, 600]$. As the value of $dl$ increased, the running time of all algorithms correspondingly increased. This observation highlights the impact of $dl$ on the overall execution time of the algorithms.

| **Algorithms** | $s = 20$ | $s = 60$ | $s = 100$ | $s = 200$ | $s = 300$ |
|---|---|---|---|---|---|
| GREEDY | $706.8 \pm 0.00$ | $706.8 \pm 0.00$ | $706.8 \pm 0.00$ | $706.8 \pm 0.00$ | $706.8 \pm 0.00$ |
| FGREEDY | $698.8 \pm 5.07$ | $704.0 \pm 2.80$ | $705.3 \pm 2.77$ | $705.8 \pm 1.58$ | $706.5 \pm 1.31$ |
| PGREEDY | $352.5 \pm 0.48$ | $352.6 \pm 0.47$ | $352.5 \pm 0.48$ | $352.6 \pm 0.48$ | $352.6 \pm 0.48$ |
| FPGREEDY | $348.1 \pm 4.11$ | $350.4 \pm 2.00$ | $351.1 \pm 1.68$ | $351.4 \pm 1.14$ | $351.5 \pm 1.00$ |

Table 2: We use FGREEDY to denote FASTGREEDY. We use PGREEDY to denote the POSTPONEDGREEDY. We use FPGREEDY to denote FASTPOSTPONEDGREEDY. Total weight of each algorithm, $s$ is the dimension after transformation. The total weight of GREEDY and POSTPONEDGREEDY don't depend on $s$. The reason why the total weight of them isn't the same in each case is there exists errors caused by different results of clustering the points. We test for $100$ times for each algorithm in each case and calculate the mean and standard deviation of the total weight. Let $A$ be the mean of the total weight of an algorithm. Let $B$ be the standard deviation of the total weight of an algorithm. In each entry, we use $A \pm B$ to denote the total weight of an algorithm when $s$ is set by a specific value.

Table 2 shows the total weight of each algorithm when $s$ changes. Since GREEDY and POSTPONEDGREEDY don't depend on $s$, the total weight of them doesn't change while $s$ changes. We can see that the total weight of each optimized algorithm is very close to that of each original algorithm respectively, which means the relative error $\epsilon$ of our algorithm is very small in practice. As $s$ increases, the difference between the total weight of each optimized algorithm and each original algorithm also becomes smaller. And the total weight of FASTGREEDY is 2 times of that of FASTPOSTPONEDGREEDY. These empirical results match our theoretical analysis. Parameter $s$ can't affect the total weight of each algorithm. The competitive ratio of FASTGREEDY and FASTPOSTPONEDGREEDY is $\frac{1-\epsilon}{2}$ and $\frac{1-\epsilon}{4}$ respectively.

**Optimal Usage of Algorithms.** The results presented above demonstrate that our algorithms exhibit significantly faster performance when the values of $n$, $d$, and $dl$ are sufficiently large. Additionally, reducing the value of $s$ leads to a decrease in the running time of our algorithms. Our proposed algorithms outperform the original algorithms, particularly when the value of $d$ is extremely large. Consequently, our algorithms are well-suited for efficiently solving problems involving a large number of nodes with substantial data entries.

**More Experiments.** To assess the impact of parameter $n$ in the presence of other variables, we varied the values of the remaining parameters and executed all four algorithms. As illustrated in Fig. 8, Fig. 9, Fig. 10, and Fig. 11, it is evident that the running time consistently increases with an increase in $n$, regardless of the values assigned to the other parameters.

To investigate the impact of parameter $s$ in the presence of other variables, we varied the values of the remaining parameters and executed all four algorithms. Fig. 12 and Fig. 13 demonstrate that the value of $s$ is independent of the original algorithms. Conversely, Fig. 14 and Fig. 15 indicate that the running time of our algorithms increases with an increase in $s$, irrespective of the values assigned to the other parameters.

To examine the impact of parameter $d$ in the presence of other variables, we varied the values of the remaining parameters and executed all four algorithms. The results depicted in Fig. 16, Fig. 17, Fig. 18 and Fig. 19 reveal that the running time of both the original and our proposed algorithms increases with an increase in $d$, regardless of the values assigned to the other parameters.

To assess the impact of parameter $\mathrm{dl}$ in the presence of other variables, we varied the values of the remaining parameters and executed all four algorithms. As evident from Fig. 20, Fig. 21, Fig. 22, and Fig. 23, the running time consistently increases with an increase in $\mathrm{dl}$, regardless of the values assigned to the other parameters.

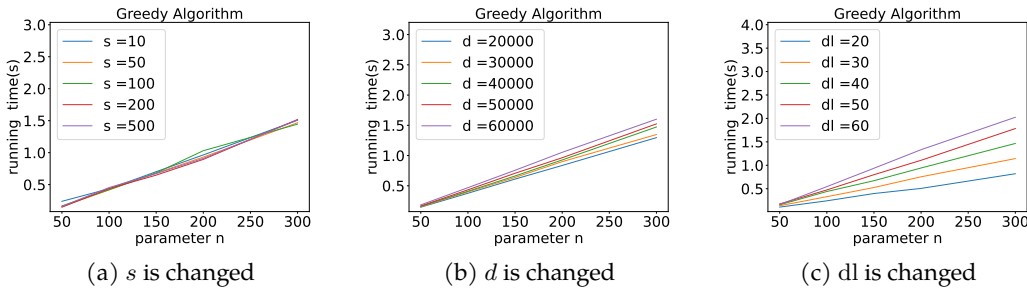

Figure 8: The relationship between running time of the greedy algorithm and parameter $n$

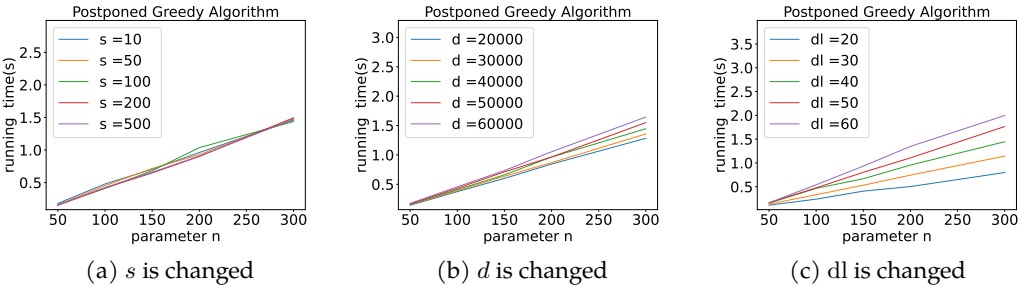

Figure 9: The relationship between running time of postponed greedy algorithm and parameter $n$

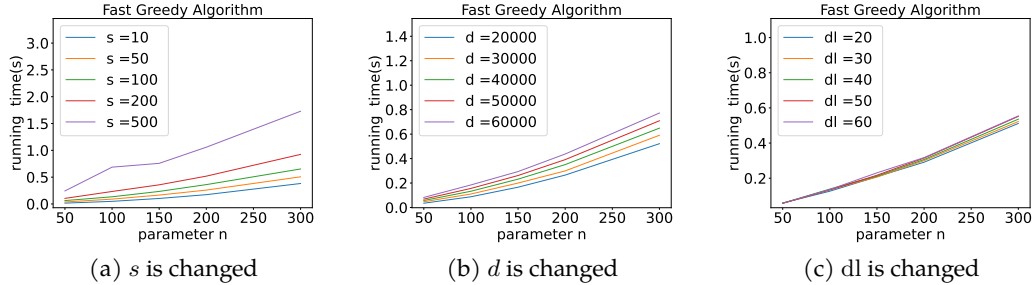

Figure 10: The relationship between running time of fast greedy algorithm and parameter $n$

## C. More Related Work

The JL lemma [45] provides strong theoretical guarantees with high probability $(1 \pm \epsilon)$-distortion, which is critical for maintaining our competitive ratio proofs. This is evidenced in Lemma 2.5 and its application in Theorems 3.4 and 3.5. Compared with principal component analysis (PCA), sketching typically requires less computational complexity $O(snd)$, where the sketching matrix is $s \times n$ and the data matrix is $n \times d$ and in our case, $s = O(\epsilon^{-2} \log(n/\delta))$ because it requires sampling of rows of the data matrix. However, PCA requires the computation of SVD, which requires $\min\{O(n^2 d), O(nd^2)\}$, for an $n \times d$ data matrix. In sketching, the value of $s$ can be adjusted, representing a trade-off between

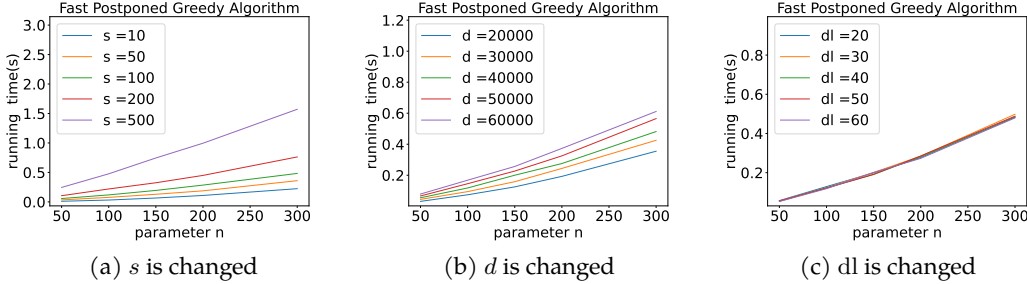

(a) $s$ is changed       (b) $d$ is changed       (c) dl is changed

Figure 11: The relationship between running time of fast postponed greedy algorithm and parameter $n$

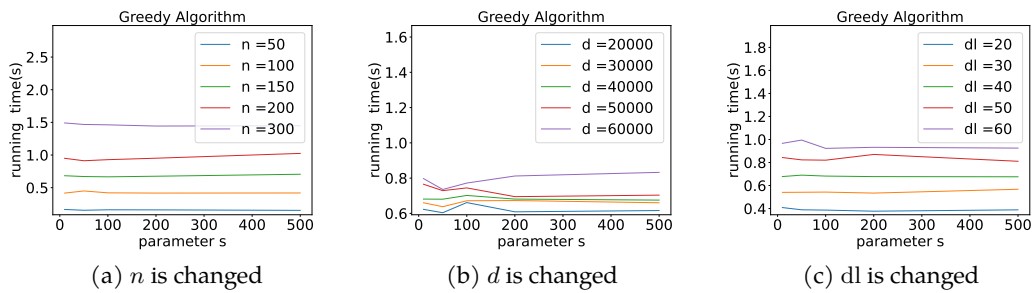

(a) $n$ is changed       (b) $d$ is changed       (c) dl is changed

Figure 12: The relationship between running time of the greedy algorithm and parameter $s$

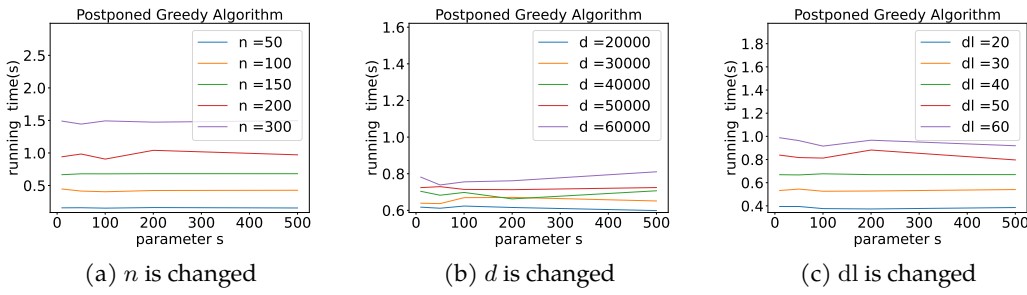

(a) $n$ is changed       (b) $d$ is changed       (c) dl is changed

Figure 13: The relationship between running time of postponed greedy algorithm and parameter $s$

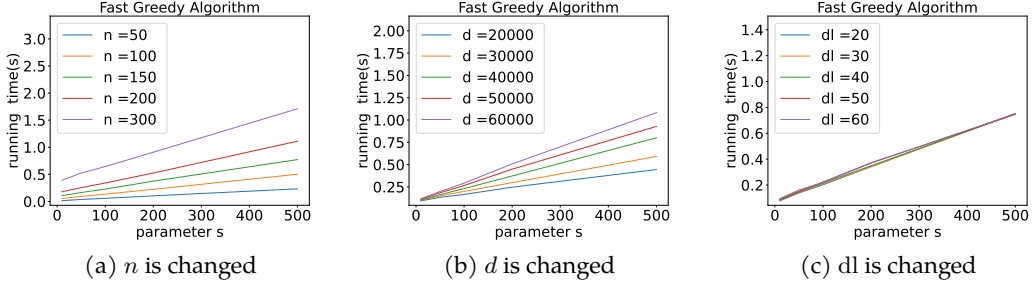

(a) $n$ is changed       (b) $d$ is changed       (c) dl is changed

Figure 14: The relationship between running time of fast greedy algorithm and parameter $s$

efficiency and accuracy. This trade-off can be also seen in our experimental results (especially in

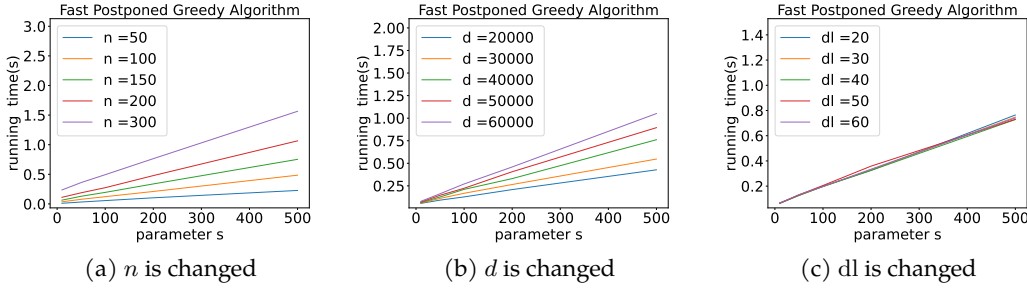

(a) $n$ is changed       (b) $d$ is changed       (c) dl is changed

Figure 15: The relationship between running time of fast postponed greedy algorithm and parameter $s$

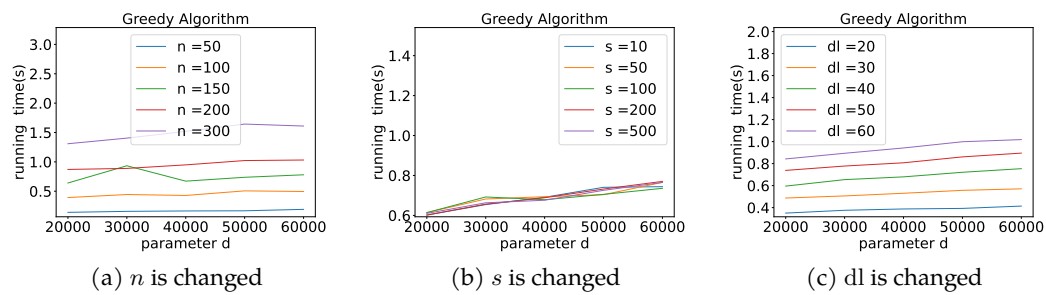

(a) $n$ is changed       (b) $s$ is changed       (c) dl is changed

Figure 16: The relationship between running time of the greedy algorithm and parameter $d$

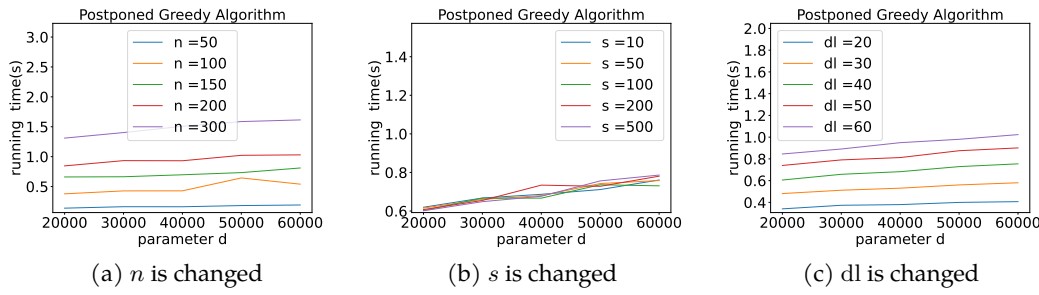

(a) $n$ is changed       (b) $s$ is changed       (c) dl is changed

Figure 17: The relationship between running time of postponed greedy algorithm and parameter $d$

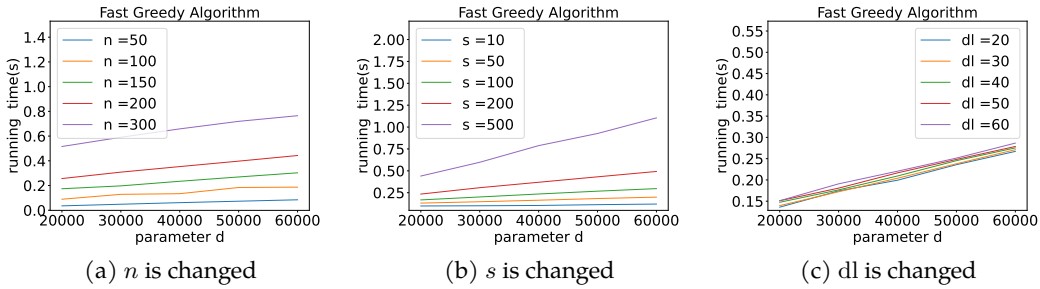

(a) $n$ is changed       (b) $s$ is changed       (c) dl is changed

Figure 18: The relationship between running time of fast greedy algorithm and parameter $d$

Table 2). Regarding accuracy, sketching can generate more random results because of its sampling procedure and the choice of $s$, whereas PCA has a more deterministic error.

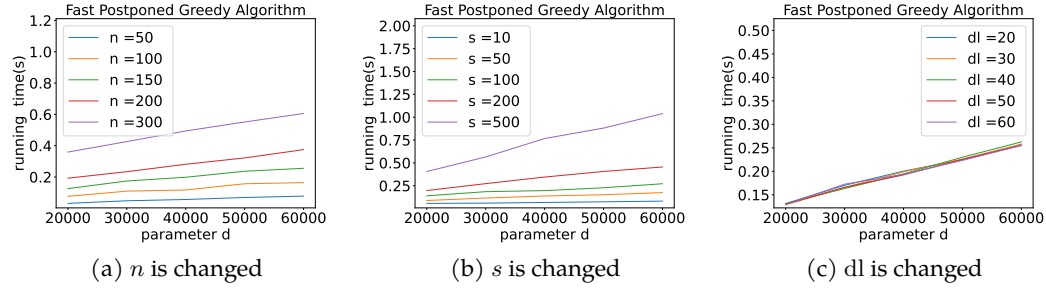

(a) $n$ is changed       (b) $s$ is changed       (c) dl is changed

Figure 19: The relationship between running time of fast postponed greedy algorithm and parameter $d$

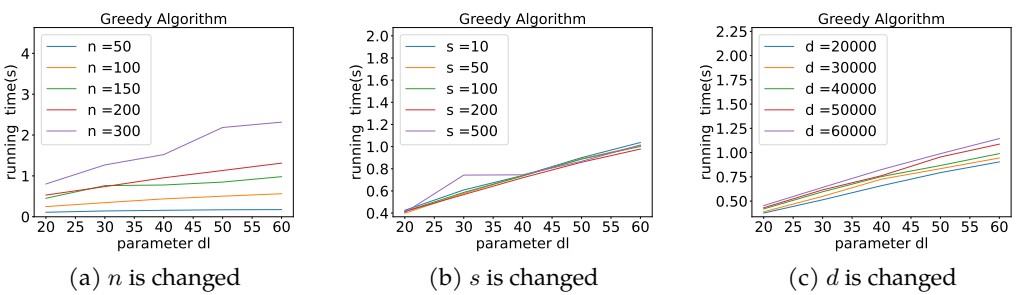

(a) $n$ is changed       (b) $s$ is changed       (c) $d$ is changed

Figure 20: The relationship between running time of greedy algorithm and parameter dl

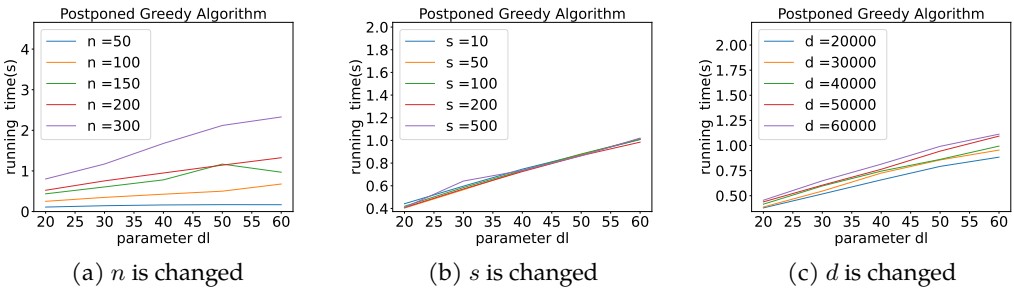

(a) $n$ is changed       (b) $s$ is changed       (c) $d$ is changed

Figure 21: The relationship between running time of postponed greedy algorithm and parameter dl

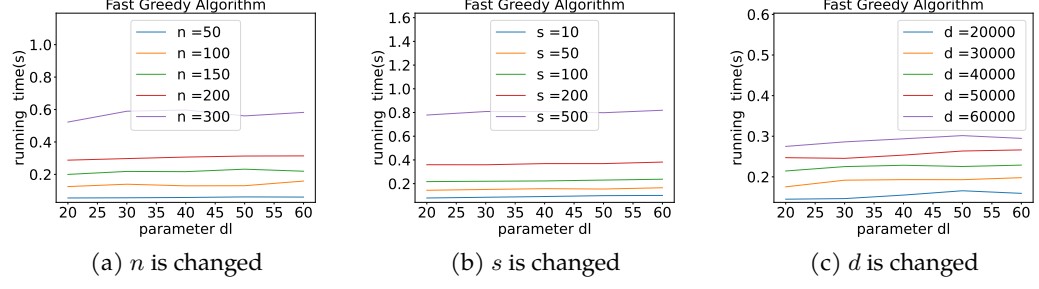

(a) $n$ is changed       (b) $s$ is changed       (c) $d$ is changed

Figure 22: The relationship between running time of fast greedy algorithm and parameter dl

More generally, other theoretical machine learning works study LLM efficiency [51–79], differential privacy [80–86], determinantal point processes [87], fast Gaussian transform [88], attack problems

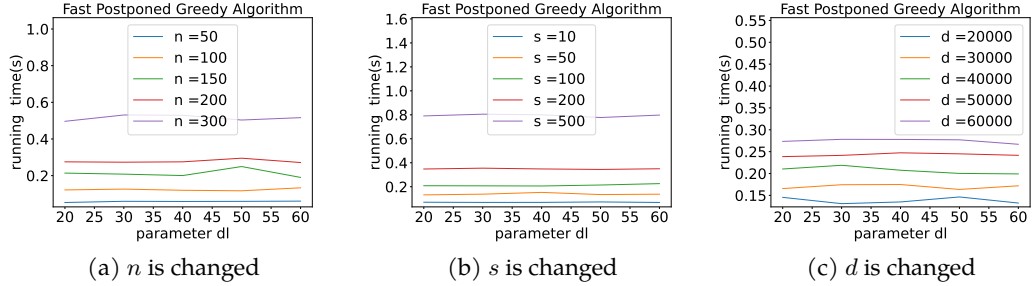

(a) $n$ is changed    (b) $s$ is changed    (c) $d$ is changed

Figure 23: The relationship between running time of fast postponed greedy algorithm and parameter dl

[89], kernel density estimation [90], online bipartite matching [89], active learning [91], leverage score [92], reinforcement learning [93–97], circuit complexity [98, 99], and fairness analysis [100].

Sketching has been applied to many other fields, such as attention approximation [101–106], $k$ means clustering [107], federated learning [108], tensor decomposition [109], weighted low rank approximation [110], linear regression [111, 112].

