# OpenReview forum: "Fast and Efficient Matching Algorithm with Deadline Instances"
_CPAL.cc/2025/Proceedings_Track — CPAL 2025 (Proceedings Track) Poster_

### Official Review · Reviewer_o9UU · 2025-01-09
**This paper discussed a fast and efficient matching algorithm which is interesting. Moreover, the authors also proposed a theoretical bound for their algorithm.**

**Rating:** 7
**Confidence:** 2

**Review:**

This paper proposes an interesting problem and also presents the theoretical bound. I have the following comments for the authors:

1. The authors claimed they reduced the bound from the original $O(nd)$ to $\tilde{O}(\epsilon^{-2}\dot (n+d))$. Could you provide more explanation for why you use a tilde on the O for your bound? looks it is uncommon for complexity. typically, we use big O or small o. You claimed $n$ is large, could you also explain the value of $d$ and the practical choice of $\epsilon$ because these two will affect your bound significantly.

2. 1.1 related work first paragraph last sentence , you claimed: Moreover, we believe... it is better to give more evidence instead of saying we believe in scientific writing.

3. 2. Preliminaries. notation $poly$ looks not defined? polynomial?

4. Lemma 2.5, $X \subset \mathbb R^d$, does $x,y\in X$? looks you do not claim this.

5. Lemma 3.1-3.3, you claimed these lemmas are informal version of Lemma A.1-A.3. However, I checked, they're identical. So? Moreover, it is not a good way to put informal version in the main paper.  you have similar issues in the following lemmas.

6.  Proof of Lemma 3.2, here you require $\epsilon\in(0,0.1)$ then looks you cannot just replace $2\epsilon$ by a new $\epsilon'$ to get your desired result since they have different ranges.

7.  Some conference names are not capitalized.

---

### Official Review · Reviewer_yyLc · 2025-01-10
**Reviews**

**Rating:** 6
**Confidence:** 2

**Review:**

The paper addresses the online weighted matching problem effectively, introducing the FastGreedy and FastPostponedGreedy algorithms with theoretical guarantees.

**Strengths:**

1. Strong mathematical rigor with clear proofs for time complexity and competitive ratios.

2. Empirical validation is conducted on both synthetic and real-world datasets, demonstrating improved efficiency over baseline algorithms.

**Weakness:**

1. The paper discusses performance on multiple datasets but lacks statistical analysis of results (e.g., error bars).

---

### Official Review · Reviewer_L9ob · 2025-01-15
**The paper is well-written and clear. In terms of content, I also believe it aligns well with the purpose of the conference. The necessary proofs for the logic are well-placed, and this helps understanding effectively.**

**Rating:** 7
**Confidence:** 2

**Review:**

The paper is well-written and clear. In terms of content, I also believe it aligns well with the purpose of the conference. The necessary proofs for the logic are well-placed, and this helps understanding effectively.

**Question:**

1. The paper uses the Johnson-Lindenstrauss lemma for dimensionality reduction. How does this approach affect the trade-off between approximation quality and computational efficiency? Are there specific scenarios where alternative dimensionality reduction techniques might offer better performance or theoretical guarantees?

2. The experimental evaluation uses both synthetic and real-world datasets. How do the algorithms perform under various arrival patterns (e.g., adversarial vs. random) and deadline distributions? Adding that discussion will make the results more solid.

3. The space complexity of the proposed algorithms is analyzed in the paper. How does this translate to practical memory usage in large-scale applications?

4. This paper proves competitive ratios for FastGreedy and FastPostponedGreedy. How do these ratios change if we consider adversarial input sequences, and are there lower bounds for this problem that suggest these ratios might be optimal? It would be beneficial to include a discussion on adversarial input sequences.

---

### Meta-Review · Area_Chair_87ST · 2025-02-02

**Recommendation:** Accept (Poster)
**Confidence:** 4

**Metareview:**

This paper proposes two new algorithms (FastGreedy and FastPostponedGreedy) for online weighted bipartite matching with deadlines, offering both theoretical guarantees and practical efficiency. Reviewers praised the clarity of the proofs, the empirical validation on synthetic and real datasets, and the significant speedup compared to prior methods. While suggestions were made to further clarify parameter choices, add more discussion on arrival patterns, and include additional statistical analysis (e.g., error bars), the overall consensus is that the paper’s contributions are valuable and merit acceptance.

---

### Decision · Program_Chairs · 2025-02-11

Accept (Poster)